# Uncovering anthocyanin diversity in potato landraces (*Solanum tuberosum* L. Phureja) using RNA-seq

**Luis Miguel Riveros-Loaiza[1,2], Nicolás Benhur-Cardona[3], Liliana Lopez-Kleine[3], Johana Carolina Soto-Sedano[4], Andrés Mauricio Pinzón[5], Teresa Mosquera-Vásquez[6], Federico Roda[2]***

**1** Área Curricular de Biotecnología, Facultad de Ciencias, Universidad Nacional de Colombia Sede Medellín, Medellín, Antioquia, Colombia, **2** Max Planck Tandem Group, Facultad de Ciencias, Universidad Nacional de Colombia Sede Bogotá, Bogotá, Colombia, **3** Departamento de Estadística, Facultad de Ciencias, Universidad Nacional de Colombia Sede Bogotá, Bogotá, Colombia, **4** Departamento de Biología, Facultad de Ciencias, Universidad Nacional de Colombia Sede Bogotá, Bogotá, Colombia, **5** Instituto de Genética, Universidad Nacional de Colombia Sede Bogotá, Bogotá, Colombia, **6** Facultad de Ciencias Agrarias, Universidad Nacional de Colombia Sede Bogotá, Bogotá, Colombia

* frodaf@unal.edu.co

**Data Availability Statement:** Raw reads from RNA-seq were deposited at the SRA archive from

## Abstract

Potato (*Solanum tuberosum* L.) is the third largest source of antioxidants in the human diet, after maize and tomato. Potato landraces have particularly diverse contents of antioxidant compounds such as anthocyanins. We used this diversity to study the evolutionary and genetic basis of anthocyanin pigmentation. Specifically, we analyzed the transcriptomes and anthocyanin content of tubers from 37 landraces with different colorations. We conducted analyses of differential expression between potatoes with different colorations and used weighted correlation network analysis to identify genes whose expression is correlated to anthocyanin content across landraces. A very significant fraction of the genes identified in these two analyses had annotations related to the flavonoid-anthocyanin biosynthetic pathway, including 18 enzymes and 5 transcription factors. Importantly, the causal genes at the D, P and R loci governing anthocyanin accumulation in potato cultivars also showed correlations to anthocyanin production in the landraces studied here. Furthermore, we found that 60% of the genes identified in our study were located within anthocyanin QTLs. Finally, we identified new candidate enzymes and transcription factors that could have driven the diversification of anthocyanins. Our results indicate that many anthocyanins biosynthetic genes were manipulated in ancestral potato breeding and can be used in future breeding programs.

## Introduction

For centuries tuber coloration has been an important trait for the breeding of native potato landraces in South America. Tuber coloration is a target for selection by native cultures of the Andes due to nutritional, aesthetic, and medicinal attributes associated to potato color. This selection has created hundreds of varieties with distinctive colors. Unfortunately, this diversity has been dramatically reduced in potato cultivars consumed at large scale worldwide [1–3].

NCBI with accession numbers SAMN24108099 to SAMN24108135.

**Funding:** Funded by the "Convenio 566 de 2014" between Universidad Nacional de Colombia and Colciencias; Colciencias Grant No 110171250437; and the project SAN Nariño number 108125-002 funded by International Development Research Centre (IDRC) and Global Affairs Canada through the Canadian International Food Security Research Fund (CIFSRF). The funders had no role in the design of the study; in the collection, analyses, or interpretation of data; in the writing of the manuscript, or in the decision to publish the results.

**Competing interests:** The authors have declared that no competing interests exist.

Introgression of genetic diversity from landraces into commercial cultivars has become a priority for breeding programs in order to ameliorate nutritional outcomes and adapt to future environmental challenges [1, 4, 5].

Diversity of pigmentation patterns in potato tubers is created by variation in the types of pigments accumulated, their concentrations in the skin or pulp of the tuber, and whether they accumulate homogeneously or heterogeneously in these tissues. Red and purple colors are associated with the accumulation of anthocyanins like peonidin and pelargonidin (red) or petunidin and delphinidin (purple).

Recently, anthocyanin biosynthesis genes have been directly targeted in potato breeding due to the antioxidant properties of anthocyanins and their potential to provide beneficial effects for human diet [6]. Ancestral breeding for potatoes with purple and red colors likely caused numerous genetic and regulatory changes in the anthocyanin biosynthetic pathway. However, these changes are largely unknow given that most studies on the genetic basis of anthocyanin variation in potatoes have been conducted in a small number of cultivars with highly contrasting anthocyanin levels [7–11].

Linkage QTL mapping and transgenesis have been combined to identify a handful of genes with a major role in anthocyanin biosynthesis. Among these, the most important are the causal genes at three major anthocyanin QTLs named the Developer (*D*), Purple (*P*) and Red (*R*) loci. The DFR enzyme at the *R* locus is responsible for the formation of red anthocyanins [7, 10, 11] while the F3´5´H enzyme is crucial to produce purple anthocyanins [9]. The AN2 transcription factor at the *D* locus induces the expression of multiple enzymes in the anthocyanin pathway, thus controlling overall anthocyanin content in tubers [10]. Recently, transcriptome analyses have expanded our understanding of anthocyanin biosynthesis in potatoes by identifying gene expression changes associated to anthocyanin signaling cascades in cultivars and mutants [12–19].

However, it is unknown if these same genes are responsible for anthocyanin accumulation in diploid landraces. Additionally, we don´t know how variation in the sequence and expression levels of these and other anthocyanin genes contribute to generating quantitative variation in pigmentation. Finally, previous studies have not used a phylogenetic framework to understand the genetic changes that accompanied anthocyanin evolution.

To begin answering these questions, we used a collection of diploid landraces with very variable colors and coloration patterns. This collection has been previously used to conduct metabolic GWAS (mGWAS) and pathway analyses of anthocyanin variation [20, 21]. These studies have identified anthocyanin QTLs in the potato genome and provided hypotheses on the genes and processes underlying these QTLs. We used this previous information to select a subset of 37 landraces representing the phenotypic and phylogenetic diversity of our collection (S1 Fig). Using this selected panel, we conducted transcriptome analyses aimed at finding correlations between gene sequence, gene expression and anthocyanin content. These analyses allowed us to better dissect the genetic and evolutionary basis of anthocyanin diversity in potato landraces. We found that tuber pigmentation in diploid landraces involves some of the same genes targeted in the recent breeding of colored cultivars. Our results also indicate that multiple enzymes and transcription factors were crucial for generating the diversity in pigmentation patterns.

## Results

### Correlation between data sources

This study integrated data from three sources to identify anthocyanin diversification genes: (1) genetic variation, (2) gene expression, and (3) anthocyanin content. We used Regularized and

Sparse Generalized Canonical Correlation Analysis (RGCCA) to evaluate the correlations between these three sources of data. RGCCA is an extension of the Canonical Correlation Analysis for the comparison of more than two datasets and more variables than individuals [22]. Our results indicate that overall gene expression variation among accessions could be largely explained by phylogenetic relationships between accessions: Paired correlations between data sources are higher between phylogeny and gene expression: 0.92 (Fig 1). The correlation between gene expression and phylogenetic distance is 0.92 (Fig 1A), and the correlation between gene expression and anthocyanin production is 0.73 (Fig 1B).

For anthocyanin concentration, the principal variance is due to cyanidin as its correlation is highest to the first canonical variable (S1 Table). This means that individuals separated horizontally in Fig 1B have highest divergence regarding anthocyanins, especially cyanidin levels (individuals CC0019 vs CC0011, Fig 1B). Moreover, RGCCA indicates that individuals like

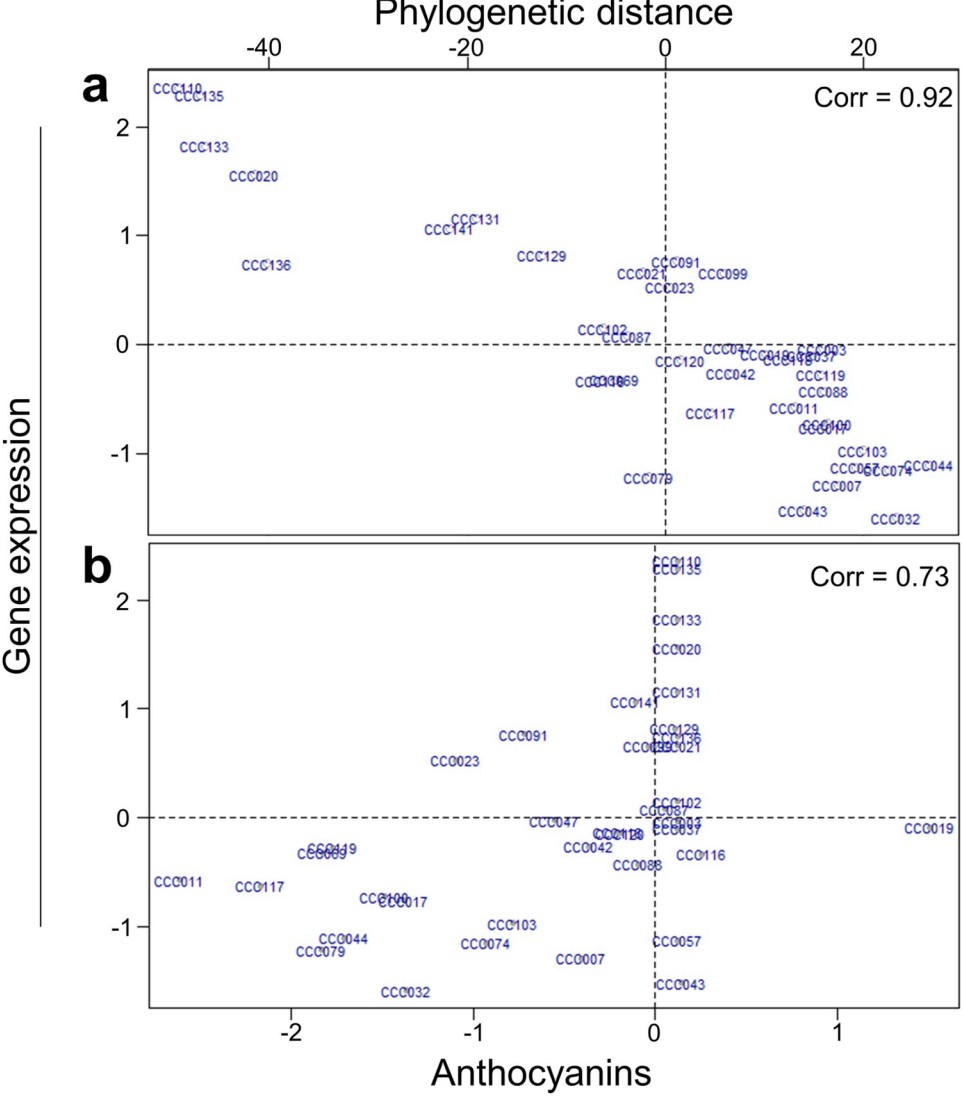

**Fig 1. Results from Regularized and Sparse Generalized Canonical Correlation Analysis (RGCCA).** The 37 potato accessions are plotted according to their RGCCA loadings for three sources of data used in our study: (**a**) global gene expression vs phylogenetic distance and (**b**) global gene expression vs anthocyanin content. Paired correlations between data sources are shown in shown in the top left.

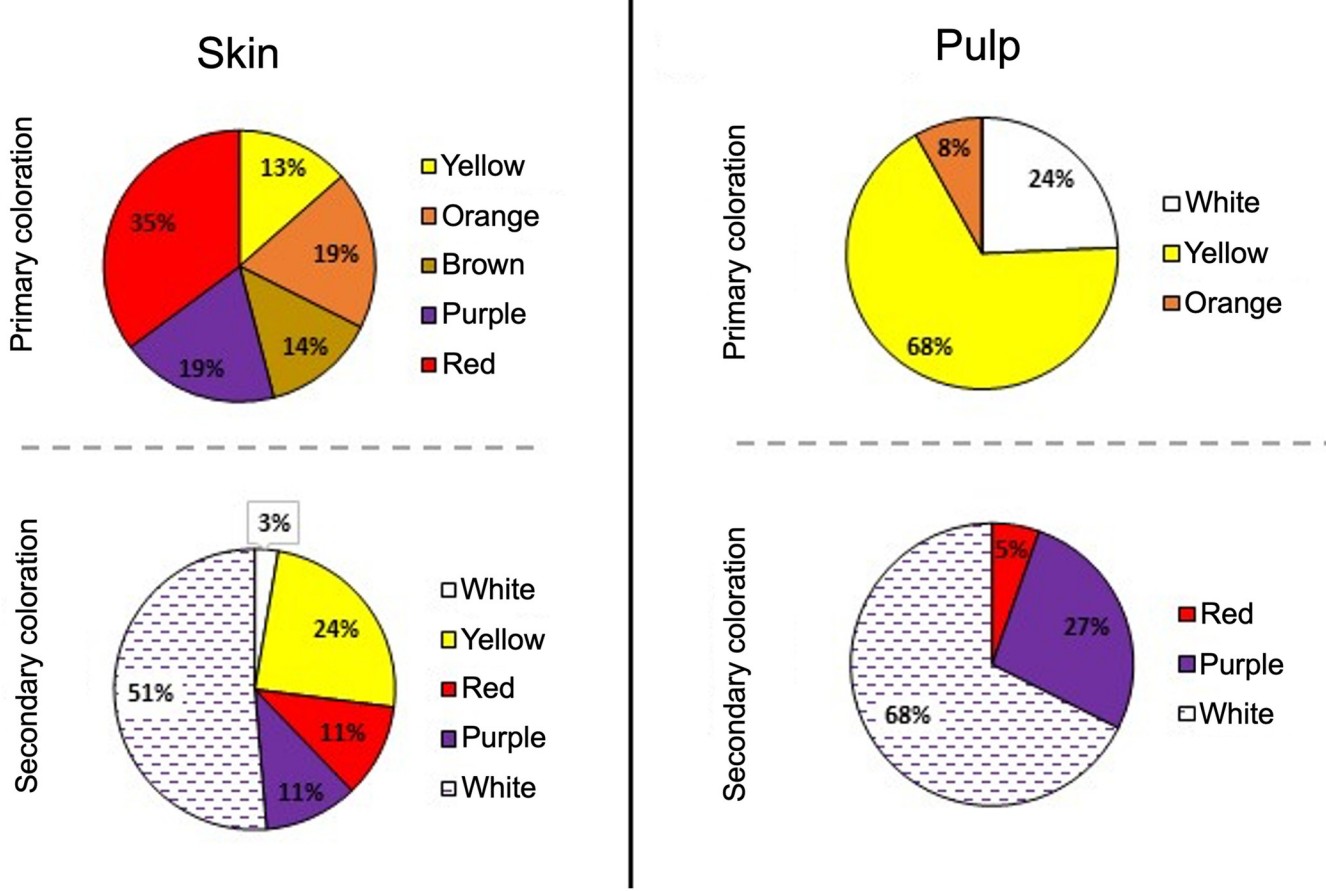

**Fig 2. Variation in tuber coloration in potato landraces.** Proportion of genotyped accessions with different primary and secondary colorations (spots) in the skin and pulp of their tubers.

CC0110, CC0135, CC0133 and CC0020 differ from other accessions mainly due to differences in gene expression and that these differences are very similar to the phylogenetic distances (Fig 1A).

### Differential expression analyses

Variation in many aspects of anthocyanin related coloration can be better described qualitatively. We categorized the coloration of tubers from our collection by defining the primary color and secondary color (i.e., spots) in the skin and pulp of the tuber. About half (54%) of potato accessions have red and purple primary skin colorations, while the rest exhibit yellow, orange, and brown colorations (Fig 2). A third of accessions (32%) presented red and purple spots in their flesh (Fig 2). To identify genes associated to the creation of this qualitative variation we performed differential expression analyses by comparing gene expression between potatoes with different coloration patterns. We conducted three pairwise comparisons (see methods), (1) purple color vs light coloration, (2) red color vs light coloration, (3) homogeneous pigmentation vs heterogeneous pigmentation. The number of differentially expressed genes (DEG) detected in each comparison is shown in Fig 3A and Table 1 (full results in S3 Table).

Most DEGs are upregulated in pigmented potatoes (purple, red or heterogeneous; Table 1). Additionally, we identified 19 DEGs that were shared between the red vs light and purple vs

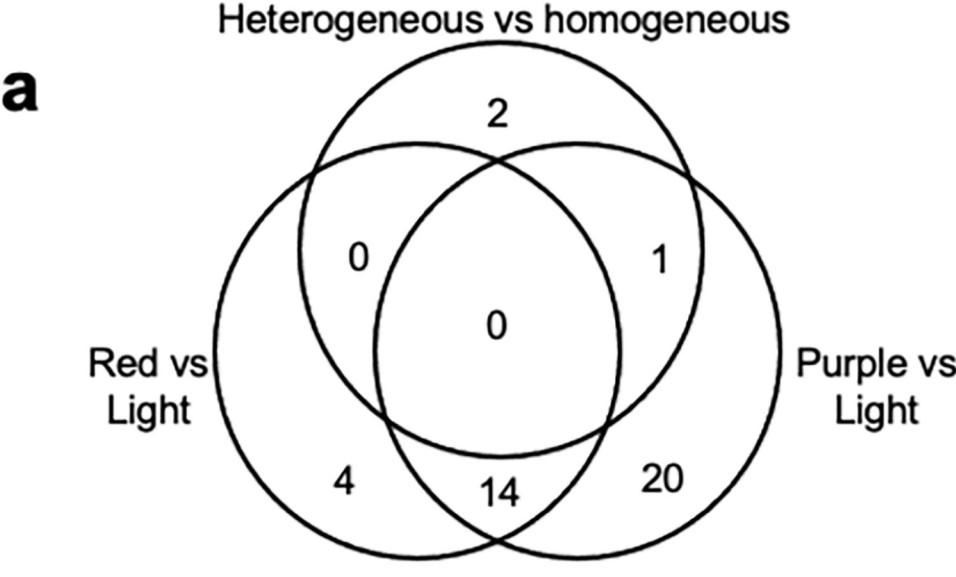

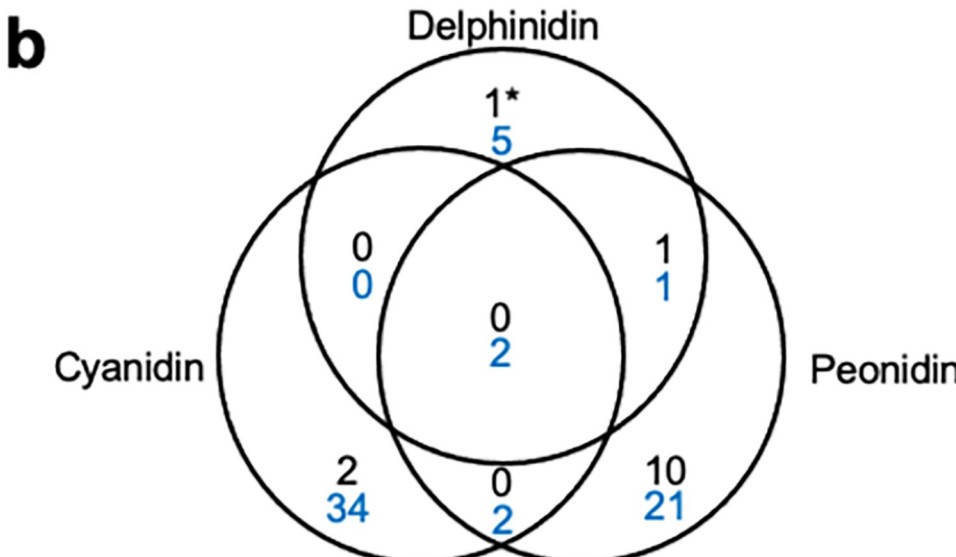

**Fig 3. Number of genes expressed differentially between accessions with different colorations and anthocyanin contents.** (**a**) Venn Diagram showing the number of differentially expressed genes (DEGs) in three comparisons between potatoes with different colorations: purple vs light, red vs light, homogeneous vs heterogeneous. (**b**) Venn Diagram showing the number of genes showing correlations between expression and the content of three anthocyanins (ExM genes): Cyanidin, delphinidin, and peonidin. The number of genes with positive correlations to anthocyanin content is shown in black; the number of genes showing negative correlations to anthocyanin content is shown in blue. The DFR gene, which is marginally associated to delphinidin is indicated with (*).

light comparisons (Figs 3A and 5B). Importantly, the causal genes at the D, P and R loci were more expressed in accessions with high anthocyanin content: StAN2 (Soltu.DM.10G020850) and F3'5'H (Soltu.DM.11G020990) are more expressed in purple tubers while DFR (Soltu.DM.02G024900) is overexpressed in red and purple potatoes.

**Table 1. Analysis of differentially expressed genes and their correlations with anthocyanins.** Summary of the Logarithm of Change in Expression (LogFC) values for the differential expression analyses, and the correlation values between the expression of these genes with the content of the anthocyanins delphinidin and peonidin. Values highlighted in black present significant p-values < 0.1.

| Gene ID v6.1 | Code Enzyme | Function | Chromosome | logFC purple vs light | logFC red vs light | logFC Heterogeneous vs Homogeneous | GS delphinidin | GS Peonidin |
|---|---|---|---|---|---|---|---|---|
| Soltu. DM.10G020850 | AN2 | Myb TF | chr10 | **4.84** | 5.90 | 3.19 | **0.64** | **0.71** |
| Soltu. DM.08G026700 | ANS | leucoanthocyanidin dioxygenase | chr08 | **4.95** | **4.06** | 1.96 | 0.50 | **0.59** |
| Soltu. DM.09G025040 | AOMT | Anthocyanin-O-methyltransferase | chr09 | **8.72** | **7.70** | 2.50 | 0.56 | **0.64** |
| Soltu. DM.09G019660 | bHLH | basic helix-loop-helix (bHLH) TF | chr09 | **5.73** | **5.03** | 2.67 | 0.53 | 0.52 |
| Soltu. DM.05G022280 | CHI | Chalcone-flavanone isomerase | chr05 | **5.86** | 5.45 | 3.05 | 0.43 | 0.50 |
| Soltu. DM.05G001950 | CHI | Chalcone-flavanone isomerase | chr05 | **3.71** | 1.52 | **3.49** | 0.28 | 0.27 |
| Soltu. DM.05G023610 | CHS | Chalcone and stilbene synthase | chr05 | **9.25** | **7.22** | 2.25 | 0.55 | **0.64** |
| Soltu. DM.09G028560 | CHS | Chalcone and stilbene synthase | chr09 | **7.49** | 5.37 | 3.41 | 0.35 | 0.47 |
| Soltu. DM.09G025360 | CYP450 | Cytochrome P450, family 72 * | chr09 | **1.23** | 1.12 | 0.59 | 0.27 | 0.16 |
| Soltu. DM.04G032930 | CYB5 | Cytochrome B5 isoform D * | chr04 | **6.66** | **5.76** | 2.51 | 0.57 | **0.65** |
| Soltu. DM.02G024900 | DFR | dihydroflavonol 4-reductase | chr02 | **6.21** | **5.85** | 2.46 | 0.51 | **0.58** |
| Soltu. DM.11G020990 | F3'5'H | Flavonoid 3'5' hydroxylase | chr11 | **7.74** | 6.12 | 2.83 | **0.51**\* | 0.57 |
| Soltu. DM.02G023850 | F3H | Flavanone 3-hydroxylase | chr02 | **7.44** | **6.77** | 2.71 | 0.56 | **0.63** |
| Soltu. DM.02G020850 | GST | Glutathione S-transferase | chr02 | **7.41** | **6.78** | 2.75 | 0.55 | **0.62** |
| Soltu. DM.03G003200 | HMGCoA | Hydroxy methylglutaryl CoA reductase | chr03 | **1.32** | 0.91 | 0.91 | 0.30 | 0.35 |
| Soltu. DM.03G018250 | MATE | Detoxifying efflux carrier * | chr03 | **7.56** | **6.44** | 2.56 | 0.57 | **0.64** |
| Soltu. DM.05G004700 | MYB | Myb TF * | chr05 | **6.23** | 4.82 | 1.92 | 0.61 | **0.61** |
| Soltu. DM.04G004510 | MYB | Myb TF * | chr04 | **0.36** | -0.39 | **5.40** | 0.36 | 0.30 |
| Soltu. DM.12G006430 | PEPCK | Phosphoenolpyruvate carboxykinase * | chr12 | **8.51** | **6.98** | 2.70 | 0.49 | 0.53 |
| Soltu. DM.12G006400 | PEPCK | Phosphoenolpyruvate carboxykinase * | chr12 | **7.80** | 5.83 | 2.82 | 0.50 | 0.52 |
| Soltu. DM.02G026690 | UGT | UDP-Glycosyltransferase | chr02 | **0.67** | **1.54** | -0.41 | 0.16 | 0.07 |
| Soltu. DM.09G017160 | UGT | UDP-Glycosyltransferase | chr09 | **6.69** | **5.77** | 3.11 | 0.53 | **0.61** |
| Soltu. DM.12G002520 | UGT | UDP-Glycosyltransferase | chr12 | **7.95** | **7.02** | 3.23 | 0.50 | 0.57 |
| Soltu. DM.10G023760 | WRKY | WRKY TF * | chr10 | 2.88 | 2.78 | 2.43 | 0.46 | 0.45 |

* These genes have not been previously associated to anthocyanin biosynthesis.

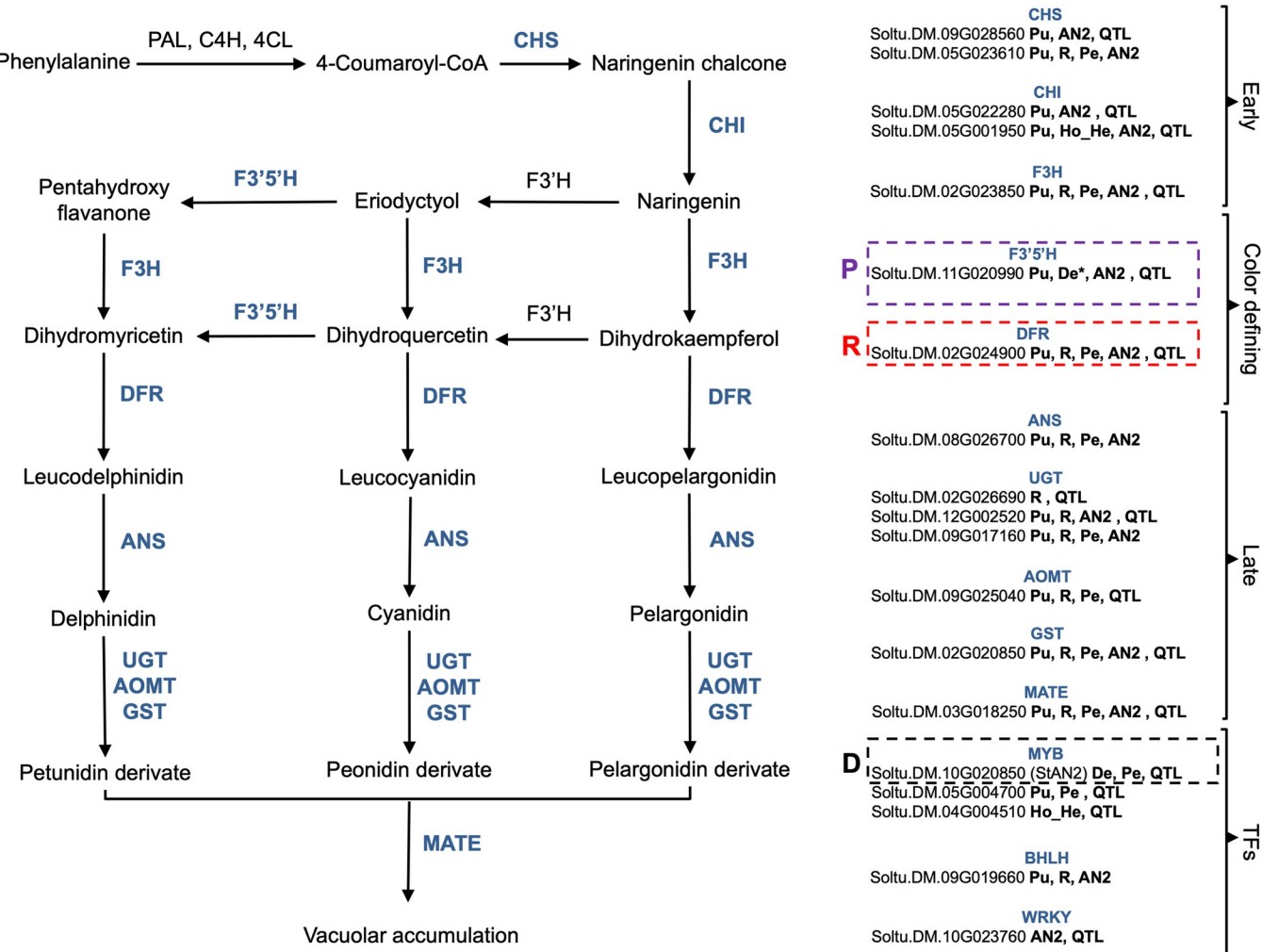

**Fig 4. DEG and ExM genes participating in anthocyanin biosynthesis.** The left panel shows schematic representation of the most important enzymatic reactions involved in anthocyanin biosynthesis. Enzymes identified among DEG or ExM genes are indicated in blue and displayed in the right panel. The right panel presents information on the different genes associated to each enzymatic function. The causal genes at the P, R and D loci are highlighted. The code of the different genes is indicated as well as significant status in different analyses: Pu: DEG purple vs light; R: DEG red vs light; De: ExM delphinidin the * indicates a marginally significant FDR = 0.11; Pe: ExM peonidin; StAN2: coexpressed with *StAN2*; QTL: located in mQTL.

## Expression by Metabolite correlation analyses

The concentration of five anthocyanins in tubers presents a continuous variation across our potato collection [20]. Therefore, an appropriate way to identify anthocyanin candidates using expression data is to search for genes whose expression is correlated to the content of each specific anthocyanin. Positive correlations occur when accessions with higher gene expression also present higher anthocyanin content. These genes are potential activators of anthocyanin production. Genes showing negative correlations are potential repressors of anthocyanin biosynthesis since accessions with higher expression at these genes also present lower anthocyanin contents.

We used WGCNA [23, 24] to calculate correlations between the expression of all genes and the content of five anthocyanins (Table 1 and S4 Table). We were thus able to identify genes whose expression shows positive correlations or negative correlations to the content in three anthocyanins, cyanidin, peonidin and delphinidin. There was relatively small overlap in genes

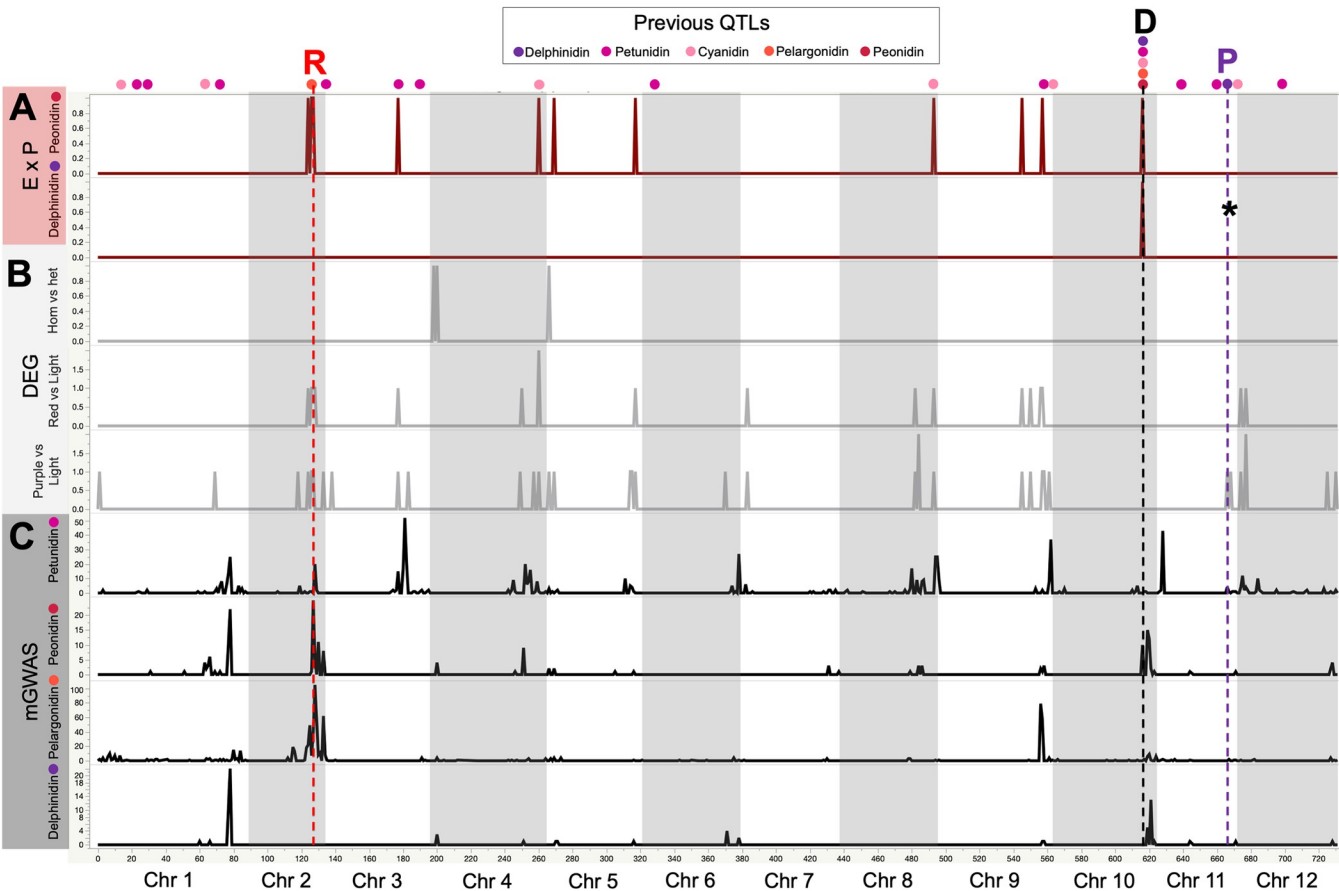

**Fig 5. Genomic location of significant genes from DEG and ExM analyses.** The genomic location across the potato genome v6.1 is presented in the X axis. Panels in the Y axis represent different analyses. (**A**) Generalized linear model (GLM) analyses of 4 anthocyanins based on genotypes extracted from transcriptomic data. For each anthocyanin we show the number of significant SNPs per 1MB interval; (**B**) Results of differential expression analyses. We show the number of genes with significantly different expression per 1MB interval for each of the three comparisons. (**C**) Results of Expression by Phenotype correlation analyses for two anthocyanins. We show the number of genes with a positive and significant correlation to each anthocyanin in 1MB intervals. The asterisk in the delphinidin lane indicates that the *F3'5'H* gene is marginally significantly correlated to delphinidin (p = 0.11). In the top of the graph, we present the location of the *R*, *D* and *P* loci. Dots represent the location of QTLs in a previous GWAS study by Parra et al (2019) [20] that used the same potato collection. The color of dots indicates which anthocyanin presented a QTL in the location.

correlated to these three anthocyanins (Figs 3B, 4 and 5A). Genes showing negative correlations were more numerous than genes showing positive correlations (Fig 3B, S4 Table). In fact, for cyanidin all significant genes showed negative correlations. Half of the genes showing positive correlations to Peonidin were also DEGs (Table 1, Fig 4).

The expression of the causal genes at the D, P and R loci was significantly correlated to anthocyanin levels. The expression of *F3'5'H* is marginally significantly correlated to variation in the purple pigment delphinidin (FDR = 0.11). On the other hand, *DFR* expression across accessions is significantly correlated to the level of the red pigment peonidin. Finally, StAN2 expression was correlated to Delphinidin and Peonidin levels.

## Functional analyses

We annotated DEG and ExM genes using the Tair and Refseq databases and searched for functions and processes enriched in these gene sets by running gene set enrichment analyses (GSEA). We found that DEG sets for the purple vs light and red vs light comparisons are

**Table 2. Gene set enrichment analyses.** Summary of significantly enriched terms in gene sets from analyses of DEG (Purple vs Light and Red vs Light) and ExM (cyanidin, delphinidin and peonidin). Terms enriched in a specific set (FDR < 0.1) are indicated with "X". ExM sets were divided between genes showing positive or negative correlations. Only sets with significantly enriched functions are presented. Complete results are presented in S2 Table.

| Term | Purple vs Light | Red vs Light | Cyanidin negative | Delphinidin negative | Peonidin negative | Peonidin positive |
|---|---|---|---|---|---|---|
| Metabolic process | | X | | | | |
| Biosynthesis of secondary metabolites | X | X | | | | X |
| Flavonoid biosynthesis | X | X | | | | X |
| Flavonoid glucuronidation | | X | | | | |
| Quercetin 3-O-glucosyltransferase activity | | X | | | | |
| UDP-glucuronosyl/UDP-glucosyltransferase | | X | | | | |
| Glycosyltransferase | | X | | | | |
| Transferase | | X | | | | |
| GPI-anchor | | | X | | | |
| Anchored component of membrane | | | X | | | |
| Integral component of membrane | | | | X | X | |
| Intracellular membrane-bounded organelle | | X | | | | |
| Membrane | | | | X | X | |
| Transmembrane | | | | X | X | |
| Transmembrane helix | | | | X | X | |
| Ribosome | | | X | | | |

highly enriched in functions associated to the production of secondary metabolites and specifically to flavonoid biosynthesis (Table 2 and S2 Table). Genes involved in flavonoid glucuronidation (ie UDP-glucuronosyl/UDP-glucosyltransferase) and intracellular membrane-bounded organelles were particularly overrepresented among DEGs.

For the analysis of ExM sets we divided each of the three gene sets (cyanidin, peonidin and delphinidin) between genes showing either positive or negative correlations to the three anthocyanins (5 final sets since no genes show positive correlations to cyanidin). Genes showing positive correlations to peonidin were enriched in functions associated to flavonoid biosynthesis and the biosynthesis of secondary metabolites. Sets of genes showing negative correlations to the three anthocyanins were enriched in terms associated to membrane localization (Table 2). We used annotations to determine the specific role of DEG and ExM genes involved in flavonoid/anthocyanin biosynthesis. We identified 18 enzymes and 5 transcription factors (Fig 4). These included the causal genes at the D, P and R loci (Fig 4). Interestingly, all these known flavonoid/anthocyanin genes are more expressed in accessions with high anthocyanin content than in accessions with low content of these compounds (Table 1).

Transcription factors identified in our study included three R3-R3 MYB TFs: the causal gene at the D locus *StAN2* (Soltu.DM.10G020850) [10] as well as two additional genes (Soltu.DM.05G004700 and Soltu.DM.04G004510). We also identified a BHLH TF named *StbHLH1* (Soltu.DM.09G019660) and a WRKY TF known as St*WRKY 13* (Soltu.DM.10G023760). Finally, we identified a SAUR-like auxin-responsive protein (Soltu.DM.01G035500).

Our DEG and ExM sets contained 18 enzymes known to participate in all stages of anthocyanin biosynthesis. These "structural" genes can be divided into five categories: (i) enzymes acting early in the flavonoid pathway, (ii) enzymes acting on the first steps of anthocyanin production, and (iii) enzymes acting in the last stages of anthocyanin production (Fig 4).

i. Five enzymes act early in the flavonoid pathway, two chalcone synthase genes (*CHS*; Soltu.DM.05G023610, Soltu.DM.09G028560), two chalcone isomerase genes (*CHI*; Soltu.DM.05G001950, Soltu.DM.05G022280), and a flavanone-3-hydroxylase (*F3H*; Soltu.

DM.02G023850). Chalcone synthase catalyzes the first committed step in the flavonoid bio-synthetic pathway, the condensation of p-coumaroyl-CoA with malonyl-CoA to generate chalcone. Chalcone is then converted to dihydrokaempferol by a series of steps mediated by *CHI* and *F3H*.

ii. Two enzymes catalyze the first steps of anthocyanin production. The dihydroflavonol reductase gene (*DFR*; Soltu.DM.02G024900) is the causal gene at the R locus and catalyzes the formation of the precursor of red anthocyanin pigments like peonidin and pelargonidin [7, 10, 11]. The flavonoid 3',5'-hydroxylase gene (*F3'5'H*; Soltu.DM.11G020990) is the causal gene at the P locus and catalyzes the formation of blue anthocyanins like delphinidin [9].

iii. Six genes mediate late stages of anthocyanin production, including an anthocyanidin synthase (*ANS*; Soltu.DM.08G026700), an anthocyanin O-methyltransferase (*AOMT*; Soltu.DM.09G025040), three UDP-glycosyltransferases (*UGTs*; Soltu.DM.09G017160, Soltu.DM.12G002520, and Soltu.DM.02G026690), and a glutathione s-transferase (*GST*, Soltu.DM.02G020850). These enzymes mediate the conversion of dihydroflavonols into colored anthocyanins. Briefly, leucoanthocyanidins are converted to anthocyanidins by ANS. Then various UGTs, AOMTs and GSTs decorate the compounds, giving each antho-cyanin its specific color.

iv. Two genes were involved in anthocyanin transport to the vacuole, a multidrug and toxic compound extrusion (*MATE*, Soltu.DM.03G018250) and the vacuolar protein *PEP3* (Soltu.DM.06G005370). *MATE* is homologous to Arabidopsis *DTX35* gene involved in anthocyanin transport to the vacuole [25–27]. PEP3 is located in an anthocyanin QTL [21] in the collection used in our study.

Many of the genes identified in our analyses have been associated to flavonoid biosynthesis. However, we also identified some genes that have not been previously associated to this path-way (Table 1). These include five enzymes: a cytochrome P450 (Soltu.DM.09G025360) a cyto-chrome B5 (Soltu.DM.04G032930), two Phosphoenolpyruvate carboxykinases (*PEPCK*, Soltu.DM.12G006430, Soltu.DM.12G006400) and a Hydroxy methylglutaryl CoA reductase (*HMGR*, Soltu.DM.03G003200). We also identified four potentially novel anthocyanin TFs—a SAUR-like auxin-responsive protein (Soltu.DM.01G035500), two MYB TFs (Soltu.DM.05G004700, Soltu.DM.04G004510), and a WRKY TF (Soltu.DM.10G023760).

## Genetic correlation analysis

We searched for expressed genes whose sequence is associated to anthocyanin variation. We extracted the genotypes of the 37 individuals by mapping the transcriptome reads to the potato genome (v6.1) and calculated associations between the genotype at 65,672 SNPs and variation in 5 anthocyanin compounds. We identified 1,574 SNPs showing significant associations to the content of 4 anthocyanins (Full results in S5 Table). No biological functions were overrep-resented in the 582 genes containing these SNPs.

We searched for significant SNPs in anthocyanin genes and found that 6 genes with antho-cyanin/flavonoid annotation contained significant SNPs: Two flavonol synthase genes (FLSs; Soltu.DM.01G030800 and Soltu.DM.02G028350), two glycosyltransferase genes (GSTs; Soltu.DM.01G047130 and Soltu.DM.12G003480), a peroxidase (Soltu.DM.04G027660), and chal-cone-flavanone isomerase gene (CHI; Soltu.DM.05G001950). The CHI gene, which contains a polymorphism associated to Peonidin variation, was also differentially expressed between potatoes with different colors and pigmentation patterns (Table 1). Additionally, a FLS gene

(Soltu.DM.02G028350) located at the R locus contains 10 SNPs significantly associated to Peonidin variation.

We analyzed the distribution of significant SNPs and found that genomic regions containing a high density of significant SNPs (i.e. QTLs; Fig 5C) were located in chromosomes 1, 2, 3, 4, 9, 8 and 10. All of these regions corresponded to QTLs identified in previous studies that used genomic variation 20,21 (Fig 5 top). We also detected two new QTLs for petunidin in Chr. 6 and 11 (Fig 5 top). Finally, we asked if DEG and ExM genes are located within QTLs from previous mGWAS (Fig 5 top, S5 Table). We found that 60% of DEG and ExM genes were in QTLs (Figs 3 and 4).

## Discussion

### Anthocyanin diversity involves enzymes at all stages of the flavonoid pathway

The genetic basis anthocyanin production has been particularly well studied in the Solanaceae family. The homologs of multiple biosynthetic genes have been identified in the potato genome [12] but their function in the creation of tuber color diversity is not well known. In this study we showed that the expression of many of these genes is correlated to tuber coloration and anthocyanin content in a diverse potato collection. In fact, flavonoid biosynthesis functions were highly overrepresented among ExM and DEG genes. This enrichment in anthocyanin functions indicates that coloration divergence involved expression changes in multiple flavonoid homologs. Gene annotation revealed that most of these flavonoid genes are enzymes or "structural" genes. These enzymes participate in all stages of anthocyanin production (Fig 4) and are more expressed in accessions with high anthocyanin concentrations (Table 1), suggesting that an increase in their expression caused an increase in anthocyanin content during landrace breeding.

Many of the enzymes identified in our study are also responsible for anthocyanin diversity in potato cultivars. Firstly, our analyses identified the *F3'5'H* and *DFR* genes mediating the production of red and purple colorations in cultivars [7, 9–11]. Consistently, we found that the *F3'5'H* at the Purple locus is overexpressed in purple landraces and is contained within a QTL for purple anthocyanins delphinidin and petunidin. Additionally, *DFR* is at the Red locus located within a QTL for the red pigments peonidin and pelargonidin and its expression is significantly correlated to the level of peonidin. These results indicate that these two enzymes were also instrumental for determining whether the tubers of landraces are purple or red. Other enzymes identified in our analyses are also to determine coloration in cultivars. For instance, all candidate genes acting early in the flavonoid pathway (*CHS*, *CHI and F3H*) are involved in anthocyanin accumulation under stress responses [15, 28]. Additionally, StANS has been cloned and shown to promote anthocyanin synthesis in tubers [29]. Finally, UGTs are differentially expressed between cultivars with different tuber colorations [30] and cause an accumulation of anthocyanins in transgenic potato [6, 31]. UGTs seem to be particularly important for anthocyanin diversification in landraces since DEG and ExM gene sets are highly enriched in functions related to Flavonoid glucuronidation.

As mentioned previously, all enzymes with flavonoid annotations show positive correlations to anthocyanin content. However, the expression of many genes is negatively correlated to anthocyanin levels. Some of these negative correlations could be the result of tradeoffs and crosstalk between biosynthetic pathways. We found that genes with negative correlations were enriched in terms associated to membrane localization, suggesting that membrane physiology is important in this crosstalk. Finally, we also identified five enzymes that have not been associated previously to anthocyanin production but could be involved in linking anthocyanin

biosynthesis to other routes of the primary metabolism (ie. PEPCK are involved in photosynthesis and sugar biosynthesis) and specialized metabolism (ie. HMGR is a rate-limiting enzyme in the synthesis of terpenoids). Future analyses on the functions of these genes are also essential for understanding anthocyanin evolution.

## The MBW complex also determines tuber coloration in landraces

The expression of anthocyanin genes is tightly controlled by the ternary complex 'MBW' [32], composed of a MYB TF, a basic helix–loop–helix (BHLH) TF and a WD40 repeat protein. In this study we identified three MYB TFs and a BHLH TF (Fig 4, Table 1). Two of these genes have been shown previously to govern anthocyanin production in potato tubers through the formation of the MBW complex. The MYB TF known as *StAN2* (Soltu.DM.10G020850) was originally identified as the causal gene at the *D* locus (the developer locus) and is responsible for tissue-specific anthocyanin accumulation in tuber skin of potato cultivars [10]. This gene is a hotspot for color evolution in plants since it regulates the expression of multiple anthocyanin biosynthetic enzymes [33, 34]. In our collection of diploid landraces, the expression of the *StAN2* gene presents the strongest positive correlation to the content of the anthocyanins delphinidin and peonidin. We also found that this gene is co-expressed with most flavonoid structural genes identified in our study. These results indicate that *StAN2* is also a master regulator of anthocyanin production in tubers of diploid potatoes.

Importantly, *StAN2* is located within a QTL associated to five anthocyanins [20, 21]. This suggests that *StAN2* pleiotropically governs the production of multiple anthocyanins. However, the genomic region harboring *StAN2* contains other known anthocyanin genes [20, 21, 35–37]. To evaluate if this multi-trait QTL is due to *StAN2* pleiotropy or to the effect of other nearby anthocyanin genes we looked at differentially expressed genes contained in this genomic region. The most significant SNPs from mGWAS are distributed over a 4MB region in Chr. 10 containing 4 DEG and ExM genes. These included the St*WRKY* 13 TF (Soltu.DM.10G023760) as well as an UGT (Soltu.DM.10G020910). The expression of *WRKY 13* was correlated to variation in peonidin and delphinidin. St*WRKY 13* activates the transcription of StCHS, StF3H, StDFR, and StANS in potato tubers [38, 39]. Altogether our results are consistent previous studies indicating that *StAN2* pleiotropically governs the production of different anthocyanins [33]. However, other linked genes like the *WRKY 13* TF and the UGT might also be contributing to anthocyanin variation.

On the other hand, we identified a BHLH, known as *StbHLH1*(Soltu.DM.09G019660), whose expression is higher in tubers with purple and blue colorations. Previous studies have shown that this gene interacts with *StAN2* in the MBW complex and is essential for anthocyanin accumulation in potato cultivars [15, 16, 34]. These results suggest that *StbHLH1* and *StAN2* also interact to govern anthocyanin production in diploid potato tubers. Notoriously, we did not detect an association to anthocyanin content for the WD40 TF *StAN11* (Soltu.DM.03G020340) which is the third member of the MBW complex [39]. Finally, we also detected two additional MYB TFs among our DEG and ExM genes. One of these genes (Soltu.DM.05G004700) is homolog to Arabidopsis *MYB3* that regulates phenylpropanoid biosynthesis [40], a crucial step in anthocyanin production. The other MYB candidate (Soltu.DM.04G004510) is overexpressed in potato tubers with heterogeneous pigmentation (i.e. spots). This gene is homologous to Arabidopsis *FLP*, which coordinates auxin related cell type differentiation [41–43]. This gene could therefore be mediating the differentiation of pigmented and unpigmented cells in the tubers with heterogeneous coloration.

Altogether our results indicate that breeding for colored potato varieties targeted master regulators of anthocyanin variation forming the MBW complex. By integrating genetic,

transcriptomic y metabolic data we validated the regulatory importance of well-known anthocyanin TFs like StAN2 and identified additional TFs that could be acting in concert with them.

## Most DEG and ExM genes are in QTLs

Two types of genes can be involved in anthocyanin diversification. Causal genes contain mutations that changed anthocyanin accumulation and were submitted to natural or artificial selection. On the other side, are genes acting downstream the causal genes, whose expression is indirectly affected by the mutations in the causal gene. These downstream genes can also be important for the phenotypic change, but they are do not contain mutations changing anthocyanin content. The gold standard experimental approach to prove the causal role of a gene in a phenotypic change is a combination of genetic mapping and transgenesis. Only a handful of causal anthocyanin genes have been identified in potato, including the *F3´5´H*, *DFR*, *StAN2*, and *StANS* genes described previously. We propose that DEG and ExM genes located within QTLs from mGWAS are good candidates to be causal genes. Consistently, we found that all the causal genes previously identified in potato cultivars were differentially expressed and contained in metabolic QTLs in our collection. Therefore, DEG and ExM genes located within QTLs are good candidates to focus future validation studies.

Recent studies have used transcriptome analyses to search for differentially expressed genes between mutants and cultivars with contrasting pigmentation patterns [12, 13, 19, 29, 44–49]. These expression analyses have helped understand the gene networks involved in anthocyanin biosynthesis. However, we ignore if differentially expressed genes caused anthocyanin diversification or they are only the result regulatory cascade triggered by a major causal gene like *StAN2*. We found that 60% of DEG and ExM genes are in QTLs and a large proportion of them are homologous to known anthocyanin/flavonoid genes. This suggests that selection for anthocyanin pigmentation in potato landraces involved mutations in multiple genes participating in all stages of the flavonoid pathway.

This relatively diverse sets of candidates contrast with studies in other plants like Petunia [33, 50–53] and Phlox [54, 55] where pigmentation evolution involved a small number of loci and specific parts of the pathway (ie mostly "branching" genes like *DFR*, *F3'5'H* and *F3'H*). We hypothesize that the relatively large number of putative causal genes in potato could be the result of a long process of selection for pigmentation diversity rather than directional selection for a specific coloration. This hypothesis is supported by the large number of coloration patterns observed in potato landraces found in indigenous communities from the Andes. The breeding process that generated this diversity possibly targeted multiple genes affecting the nature, concentration, and spatial distribution of pigments in the tubers.

The mutations responsible for evolutionary change can modify the expression of genes or affect their coding sequence and biochemical function. Regulatory mutations are characterized by causing allele specific differential expression while functional mutations are characterized by causing changes in the sequence of a protein. We found that most expressed flavonoid genes located within QTLs did not contain polymorphisms associated to anthocyanin variation (except for *CHI* and *FLS*). This suggests that regulatory mutations might be largely responsible for anthocyanin variation. Anthocyanin evolution in plants often involves regulatory changes given that these pigments are essential part of defenses [55–57]. Additionally, regulatory changes are preeminent in crop domestication, where subtle alterations in expression are favored to avoid pleiotropic effects [58–61]. Consistently, previous studies of StAN2 [10, 37] and DFR [7, 11] showed that regulatory mutations in these genes were likely responsible for the evolution of potato coloration in cultivars.

Functional mutations can cause changes in the phenotype of an organism without changes in the expression of the gene where the mutation occurred. We identified five flavonoid

enzymes containing SNPs showing genotypic associations to anthocyanin content. These genes are candidates to contain functional mutations that were targeted by breeding. The most interesting candidate is the CHI gene (Soltu.DM.05G001950), which is differentially expressed and contains a polymorphism associated to Peonidin variation. Interestingly, we also identified flavonoid enzymes that were not differentially expressed across accessions but contained putatively functional mutations. For instance, a FLS gene located in the R locus (Soltu. DM.02G028350) contained 10 significant SNPs in genotypic association analyses. FLS catalyzes de first step in the production of flavonols, diverting precursors from the anthocyanin pathway. Allelic variation in FLS genes is associated to anthocyanin variation in members of the Solanaceae family [33, 62]. Therefore, FLS could also be a causal gene at the R locus. These results show that transcriptome analysis is a cost-effective method to trace the evolution of biosynthetic pathways.

## Materials and methods

### Plant material

We used and initial panel of 94 diploid potato accessions of the *Phureja* group from the working collection of the Potato Breeding Program of the Universidad Nacional de Colombia. The content of five anthocyanins (delphinidin, cyanidin, petunidin, pelargonidin and peonidin) was previously measured in tubers of this collection using Ultra High-Performance Liquid Chromatographic analysis (UHPLC) [20]. Additionally the collection was genotyped with genotyping by sequencing [63] (S1 Table).

Each accession was sown in triplicate in individual bags with soil from the rural area of Suba, using a completely randomized design. Plant growth was carried in open area in the Universidad Nacional de Colombia campus, with an average temperature of 15˚C ± 5˚C, relative humidity of 70% ±10%. The time from sowing to harvest was from February to June 2020.

### Morphological description

We evaluated variation for color traits associated with anthocyanin content. Once harvested, potato tubers were cut in half and photographed. These photographs were used to characterize qualitative traits using the morphological descriptors of Huáman [64]. The parameters evaluated were: (i) primary coloration of the tuber skin, (ii) secondary coloration of the skin, (iii) primary coloration of the tuber inner flesh, and (iv) secondary coloration of the tuber inner flesh. This characterization was used for the selection of individuals to genotype and to establish categories in differential expression analysis (S1 Table).

### Selection of material, RNA extraction and sequencing

For RNA-seq we selected 37 accessions from the full collection of 94 accessions used in previous studies. We selected potato accessions representing the phylogenetic and phenotypic variation observed in the original collection. For this selection we used previously published phylogenetic analyses [21] and anthocyanin measurements [20], as well as the morphological traits described in the previous section. We selected accessions that (1) were in the different phylogenetic clades and (2) represented the range of phenotypic variation observed in each clade (S1 Fig, S1 Table).

For each of the 37 genotypes selected trough these criteria, the healthiest plant of the three replicates was chosen. For this plant, we selected a healthy and mature tuber with a diameter between 3 to 5 cm.

Tubers were quickly cut, placed in liquid nitrogen, and transported to the laboratory of the Max Planck Tandem Group at the Universidad Nacional de Colombia where they were stored at -80 C. Equal parts of tuber skin and pulp were sampled from each genotype for RNA extraction. The tissue was ground with a TissueLyser II (Qiagen, Hilden, Germany) and RNA extraction was conducted using the Qiagen RNeasy® Plant Mini Kit following the manufacturer's instructions. On-column digestion of residual genomic DNA was performed using the Qiagen RNase-Free DNase set. The purity of the samples was evaluated using the Thermo Fisher® NanoDrop (ND-ONE-W). Quantification was performed using Qubit 2.0 from invitrogen®. Finally, to evaluate the quality and integrity of the extractions, a 2% agarose gel electrophoresis was performed.

Subsequently, we only sequenced one transcriptome per accession because the goal of our study is to analyze a large number of accessions and we conducted statistical analyses that do not require replication per accession. RNA extractions were shipped to Novogene (California) for sequencing. Sequencing libraries were prepared by poly-A enrichment and using 250–300 bp inserts. These libraries were sequenced with Illumina NovaSeq™ 6000 S4 Sequencing System (paired end, read length = 150 bp). More than 6Gb of sequence were obtained for all samples and were stored in fastq files.

## Cleaning, mapping, and quality analysis of sequences

The nf-core/rnaseq pipeline version 3.0 [65] was applied to raw reads obtained from Novogene. Sequence processing pipeline was divided into three stages: read filtering and cleaning, read mapping, and quantification of transcripts. For sequence pre-treatment, raw data quality was assessed with the FastQC software [66] and read trimming was performed using the Trim-Galore tool [67] to remove adapter sequences and short, low-quality sequences.

Read mapping began with an alignment of pre-processed and cleaned read sequences to the latest potato reference genome [68] v6.1, available at http://spuddb.uga.edu/ indexed and aligned using the HISAT2 program [69]. The alignments were then sorted and indexed using SAMtools [70]. Finally, the prepDE.py3 script from the StringTie package [71] was used to quantify the number of reads per gene. These read counts were used as input for differential expression analyses as described below. Quality control of alignment, was performed by means of the Qualimap application [72] over all generated BAM files. Analyses were ran using the computational infrastructure of the Bioinformatics and Computational Systems Biology Research Group.

## Regularized and Sparse Generalized Canonical Correlation Analysis

We used Regularized and Sparse Generalized Canonical Correlation Analysis (RGCCA) to analyze the overall correlation between three quantitative data sources all available for the 37 individuals of interest were integrated for analysis: 1) Measurements of the concentration of five anthocyanins (X1); 2) Phylogenetic distances in form of the first 3 principal components of a PCA summarizing the distance dendrogram between individuals obtained from a previous study [21] and representing 20% of variance (X2) and 3) RNA sequencing obtained gene expression data on 32,923 genes transformed to TPM values (X3, S1 Table). Pre-processing of gene expression data: Descriptive statistics such as boxplot, histograms, and correlation plots for all 37 samples were done to identify aberrant samples or genes (S2 Fig). Genes with lower values than -1 in the sum of median log2 transformed TPM values were filtered out to obtain a final RNA sequencing data set of 19949 genes with information for all 37 individuals, which were normalized, and variance stabilized using the *vsn* method [73] (S3 and S4 Figs). This

dataset was further summarized through the first seven principal components of a PCA representing 56% of variance (S4 and S5 Figs).

Assessing correlation between three data sources: Regularized and Sparse Generalized Canonical Correlation Analysis (RGCCA) was conducted with the RGCCA library in R [22]. This analysis depends on a previous step of identifying a shrinkage constant, which has been done applying the method of Schäfer & Strimmer [74] in the R corpcor package. For the RGCCA the best number of components and factorial scheme (Factorial and Centroid) to be used was identified based on the highest mean retained variance (AVE). X1. After comparing AVE for the three factorial schemes (Horst, Factorial and Centroid), the factorial scheme was chosen for the final RGCCA. The correlation between pair of datasets (i.e. X 1and X2) was obtained through the square root of the eigenvalues of the correlation matrix R [75]: Shrinkage constants for the three data sets were: 0 for gene expression (X3), 1 phylogeny (X2) and 0.1252 for anthocyanin production (X3 and X1).

## Correspondence between genome versions

In this study we used the last version of the potato genome (v6.1.1) but previous research on anthocyanin variation was based on the older versions of the genome (v4.3/4.04). We therefore cross-referenced the two genomes using two types of data that were important for our analyses: Gene identity and SNP positions.

**Gene identity.** We compared the transcriptomes available at the SPUD.DB webpage (http://spuddb.uga.edu/) using the blastn command of the BLAST v2.12 software [76]. Reciprocal best hits showing a percentage identity greater than 96% across all their length were considered as belonging to the same gene. We were able to identify the correspondence for 24,758 genes (S6 Table).

**SNP position.** To compare our results with previous mGWAS of this potato collection we located previously identified SNPs [63] on the DM v6.1.1 assembly by mapping the SNPs flanking sequences. We extracted 100 nt sequences flanking each of the 87,656 SNPs using gffread [77] on the v4.3/4.04 genome. Each 100 nt flanking sequence was then mapped with Vmatch v2.3.1 (http://www.vmatch.de) allowing at most 5 mismatches and selected the top scoring alignment reported. If multiple alignments had the same top score the SNP is marked as multimapping. We were thus able to map 83,155 SNPs (S7 Table). Finally we assigned SNPs to genes using Bedtools v2.30.0 [78] using the intersect command with the annotation.gff file available at SPUD.DB (S7 Table).

## Differential expression analyses

We used the morphological categorization of potato tubers to assign accessions to different categories based in two criteria, tuber color and coloration distribution. These categories were defined based on the primary color of tuber skin and the secondary color of tuber pulp and are presented in S1 Table. For tuber coloration we defined three color categories, red, purple, and light. Accessions in the red and purple categories had red and purple colorations in the skin and pulp. Accessions in the light category had either white, yellow, orange or brown colorations. We selected 10 different accessions in each of the three-color categories. For pigmentation distribution in the tuber pulp, we defined two categories, homogenous or heterogenous pigmentation. We selected 9 different accessions in the homogeneous and heterogeneous categories.

We used the EdgeR package [79] from the R software (R v4.0.4) to conduct differential expression analyses using as input raw read counts obtained with the nf-core pipeline. Genes unexpressed or very lowly expressed were filtered using a cutoff based on the median

log2-transformed counts per gene per million mapped reads (cpm) of -1. Gene counts were normalized using the trimmed means of m values method (TMM) [80] and the variance of read counts per gene was modeled using the Poisson distribution. The filtered and normalized matrix contained expression from 19,949 genes. Finally, we tested for differential expression in paired comparisons between groups of accessions with different pigmentation patterns. We conducted three comparisons based in potato categories defined previously: (1) Red vs Light, (2) Purple vs Light, (3) Homogeneous vs Heterogeneous. Genes showing an FDR-corrected p-value lower than 0.1 were considered significantly differentially expressed for a comparison [81]. Genes showing a significant p-value in any of the three pairwise comparisons were termed DEG (ie differentially expressed genes) genes.

## Expression by metabolite correlation analyses

We searched for genes whose expression is correlated to anthocyanin levels across accessions of the collection. Briefly, we used the WGCNA package from the R software [23] to evaluate correlations the expression of 19,949 genes and the content of five anthocyanins. The Pearson correlation function (*cor.test*) was used to determine the significance of the correlations. The resulting p-values were corrected by the false discovery rate using the fdr correction of the "*p.adjust*" function from the R software. We used as input filtered and normalized gene expression data obtained with EdgeR as well as previously reported levels of 5 anthocyanins [20]. Full results are in S4 and S5 Tables.

Genes showing a significant association to any anthocyanin in correlation analyses were termed ExM (ie expression by metabolite) genes.

## Genotypic correlations

To identify polymorphisms in expressed genes that could be explaining variation in anthocyanin content we extracted genotypes from transcriptome data and then used these genotypes to calculate associations between genotype and anthocyanin levels. For genotype scoring we merged the vcf files generated through the nf-core pipeline [65] using the bcftools software v1.12 [81] and the "merge" command. The "filter" command from bcftools was used to filter variants genotyped in all accessions using a minimum quality of 20 and a minimum depth of 4 reads. We extracted genotypes from the combined vcf file with the Genome Analysis Toolkit (GATK) software v4.2.0.0 [82] using the command "VariantsToTable". By these means we identified 65,672 biallelic SNPs located in 8,938 genes.

The TASSEL v5 software [83] was used to calculate associations between genotypes at all biallelic SNPs and the content of five anthocyanins. We used a generalized linear model (GLM) and corrected for population structure with a PCA (covariance method, 5 components). The resulting p-values were corrected by the false discovery rate using the fdr method of the "*p.adjust*" function from the R software [84]. An association was considered significant if the fdr corrected p-value was lower than 0.1 [85]. We then identified genomic regions associated to a trait (i.e. QTLs) by calculating the sum of significant SNPs in non-overlapping 1 Mb intervals. Intervals containing significant SNPs were considered QTLs. Full results are in S5 Table.

## Gene annotation and enrichment analyses

To identify biological pathways that are overrepresented among genes from DEG and ExM analyses we used the Blastp algorithm from the BLAST v2.12 software [76] to compare the most recent potato proteome from SPUD.DB (High confidence representative gene models) to two annotated proteomes: (1) the Arabidopsis thaliana proteome deposited in the TAIR

database and (2) the potato proteome deposited in the Refseq database. Specifically, we assigned the best blastp hit in each of the two databases using a minimum e-value of $10^{-6}$. We also conducted gene set enrichment analyses for DEG and ExM gene sets using the "functional annotation clustering" tool at the David server (https://david.ncifcrf.gov/). We used as "gene lists" the TAIR and Refseq codes assigned to significant genes for each pairwise comparison (DEG genes) and anthocyanin (ExM) genes. We used as "background" the TAIR and Refseq codes assigned to all genes whose expression was evaluated. Terms showing an FDR-corrected p-value lower than 0.1 were considered significant. Annotation results are provided in S6 Table.

## Conclusions

Our study integrated omics to decipher some of the genetic and regulatory changes that accompanied the breeding of potatoes with different colorations. We found that tuber pigmentation in diploid landraces involves some of the same genes targeted in the recent breeding of colored cultivars. It is likely that the same alleles at these genes were repeatedly selected during the history of potatoes. Our results also indicate that multiple structural and regulatory genes were crucial for generating the diversity in pigmentation patterns that we observe across lineages. Furthermore, it is likely that genes contribute differently to anthocyanin variation across different lineages given that population stratification largely determines gene expression according to RGCCA. This research highlights the value of studying and preserving native crop diversity. Transcriptomic data generated in this study can be combined with other phenotypic measurements conducted in our collection to better understand the historical and physiological tradeoffs involved in trait evolution. Additionally, this knowledge can be harnessed for potato breeding trough genome editing and genomic selection. For instance, significant SNPs from mGWAS located nearby ExM genes could be selected to calculate genomic predictions in genomic selection breeding programs. Additionally, phylogenomic data can be used to determine which ExM genes could be targeted trough transgenesis to improve a specific potato lineage from our collection. Finally, genes whose expression is negatively correlated to anthocyanin content could be good targets for gene silencing.

## Supporting information

**S1 Fig. Phylogenetic and phenotypic variation in the potato population.** (A) Neighbor joining phylogeny conducted with all SNPs used in GWAS. (B) Anthocyanin content: Darker colorations are associated to higher values of the sum of Delphinidin, Cyanidin, Petunidin, Pelargonidin and Peonidin. (C) Samples used in transcriptome sequencing are indicated with X. (D) DEG categories for color: Red = Red; Purple = Purple, Yellow = Light; Light grey = Not used. (E) DEG categories for coloration pattern: Black = Heterogenous; Dark grey = Heterogenous; Light grey = Not used.
(TIFF)

**S2 Fig. Boxplot with information for all 37 individuals.** This information was normalized and variance stabilized using the vsn method.
(TIFF)

**S3 Fig. Correlation between the 37 samples using the VSN method.**
(TIFF)

**S4 Fig. Mean Standard Deviation plot obtained sing VSN method.**
(TIFF)

**S5 Fig. First two components of PCA applied to the gene expression data.**
(TIFF)

**S1 Table. Phenotypic and morphological variation in the population studied.** Anthocyanin content, tuber skin and interior coloration, and phylogenetic PCA.
(XLSX)

**S2 Table. Gene set enrichment analysis (GSEA).**
(XLSX)

**S3 Table. Results from differential expression analyses.**
(XLSX)

**S4 Table. Results from correlation analyses.**
(XLSX)

**S5 Table. Results from analyses of genotype by metabolism correlations.**
(XLSX)

**S6 Table. Annotation of genes from the potato genome v6.1.**
(XLSX)

**S7 Table. Polymorphisms used to run metabolic GWAS (mGWAS).**
(XLSX)

## Acknowledgments

We thank Julio Acevedo and the Potato Breeding Program of the Universidad Nacional de Colombia to provide the tubers for the experiments. We thank Prof. Maria C. Delgado as well as the students, Lady D. Gómez, Gina P. Sierra, and Pablo A. Perez for the help with plant rearing, collection, and RNA processing.

## Author Contributions

**Conceptualization:** Luis Miguel Riveros-Loaiza, Liliana Lopez-Kleine, Johana Carolina Soto-Sedano, Teresa Mosquera-Vásquez, Federico Roda.

**Data curation:** Andrés Mauricio Pinzón.

**Formal analysis:** Luis Miguel Riveros-Loaiza, Nicolás Benhur-Cardona, Liliana Lopez-Kleine, Federico Roda.

**Investigation:** Luis Miguel Riveros-Loaiza, Federico Roda.

**Methodology:** Luis Miguel Riveros-Loaiza, Federico Roda.

**Project administration:** Federico Roda.

**Resources:** Luis Miguel Riveros-Loaiza.

**Writing – original draft:** Luis Miguel Riveros-Loaiza, Liliana Lopez-Kleine, Federico Roda.

**Writing – review & editing:** Luis Miguel Riveros-Loaiza, Liliana Lopez-Kleine, Johana Carolina Soto-Sedano, Teresa Mosquera-Vásquez, Federico Roda.

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
