## [Decision Letter · Decision Letter 0]

18 Jul 2022

PONE-D-22-16485The contribution of gene expression to anthocyanin diversity in potato landraces (Solanum tuberosum L. Phureja)PLOS ONE

Dear Dr. Roda,

Thank you for submitting your manuscript to PLOS ONE. After careful consideration, we feel that it has merit but does not fully meet PLOS ONE’s publication criteria as it currently stands. Therefore, we invite you to submit a revised version of the manuscript that addresses the points raised during the review process.

We look forward to receiving your revised manuscript.

Kind regards,

Pankaj Bhardwaj, Ph.D.

Academic Editor

PLOS ONE

Journal Requirements:

We also thank the funded by the “Convenio 566 de 2014” between Universidad Nacional de Colombia and Colciecias; Colciencias Grant No 110171250437; and the project SAN Nariño number 108125-002 funded by International Deve-opment Research Centre (IDRC) and Global Affairs Canada through the Canadian International Food Security Re-search Fund (CIFSRF).

 Funded by the “Convenio 566 de 2014” between Universidad Nacional de Colombia and Colciecias; Colciencias Grant No 110171250437; and the project SAN Nariño number 108125-002 funded by International Deveopment Re-search Centre (IDRC) and Global Affairs Canada through the Canadian International Food Security Research Fund (CIFSRF).

The authors declare no conflict of interest. The funders had no role in the design of the study; in the collection, analyses, or interpretation of data; in the writing of the manuscript, or in the decision to publish the results.

Reviewers' comments:

Reviewer's Responses to Questions

**Comments to the Author**

1. Is the manuscript technically sound, and do the data support the conclusions?

Reviewer #1: Partly

Reviewer #2: Yes

2. Has the statistical analysis been performed appropriately and rigorously? 

Reviewer #1: Yes

Reviewer #2: Yes

3. Have the authors made all data underlying the findings in their manuscript fully available?

Reviewer #1: Yes

Reviewer #2: Yes

4. Is the manuscript presented in an intelligible fashion and written in standard English?

Reviewer #1: Yes

Reviewer #2: Yes

5. Review Comments to the Author

Reviewer #1: Differential expression between potatoes with different colorations were analyzed and genes related to the flavonoid-anthocyanin biosynthetic pathway were identified by authors. It’s important for the genetic understanding of anthocyanin pigmentation in potato. The manuscript, however, should be revised with more concise words.

Other Comments for Manuscript:

1. Line 69-80, It would be better to be concise.

2. Line 85-86, “RGCCA” instead of “Regularized and Sparse Gener-alized Canonical Correlation Analysis (RGCCA)”.

4. Line 94, “shown in shown in the top left”, twice? In Fig 1.tiff, Smaller font of the name of the potato accessions would be better.

5. Line 141, Significant level for p-values is 0.01 or 0.05?

6. Line 227, In Fig 5, it would be better to scale in the X axis for each chromosome individually, instead of in an accumulated scale for all 12 chromosomes.

7. Line 344, “were likely responsible for” instead of “where likely responsible for”?

8. Line 375-397, The two sections, “Selection of material to be sequenced” and “RNA extraction and sequencing”, could be merged. Anyway, the sequencing methods could be described in a brief way.

9. Line 383, 37 (even 94) accessions are not a large number of accessions, considering the sequencing cost. How the statistical analyses support “one transcriptome per accession”?

10. The genetic backgrounds of 37 landraces are different, even though all these landraces belong to Phureja group. Did the difference interfere the Differential Expression Analyses? It should be discussed in the manuscript.

11. The quality of the written English is not satisfying. There are spelling and vocabulary mistakes needing corrections in the manuscript. Such as “an-thocyanin” in abstract. References in the text need to be enclose in brackets, such as line 378.

12. There are two names of the MYB gene (Soltu. DM.10G020850), StAN1 and StAN2.

13.The differential genes analyzed were not verified by any experiments, such as qPCR and dual luciferase reporter assay.

14. The WRKY13 gene in this manuscript has been reported and should be included in the discussion. StWRKY13 promotes anthocyanin biosynthesis in potato (Solanum tuberosum) tubers. Functional Plant Biology, 2022, 49(1):102-114

14. “our results are summarized in Fig 4” in line 167 should be deleted.

Reviewer #2: The manuscript “The contribution of gene expression to anthocyanin diversity in potato landraces (Solanum tuberosum L. Phureja)” is well presented. This can be accepted after addressing minor comments as below:

Title: Modify it, something like: ‘Uncover anthocyanin diversity in potato landraces applying RNA-seq.......approaches’

Abstract: Line 26, modify it, or mention first and second crop. Since there are several other vegetables rich in antioxidants like tomato etc.

Introduction: Line 40-41, modify, not good sentence. Modify from L40 to 64, in a story format. Its a bit scattered.

Results: L89-90: Besides two, mention the correlation between phylogeny and anthocyanin production as well.

L102: (ie spots) ?? Correct ‘i.e.’ spots

Check for typo and grammatical errors in throughout the manuscript.

6. PLOS authors have the option to publish the peer review history of their article (what does this mean?). If published, this will include your full peer review and any attached files.

Reviewer #1: No

Reviewer #2: **Yes: **Jagesh Kumar Tiwari

---

## [Author Response · Author response to Decision Letter 0]

12 Aug 2022

Detailed responses to reviewer # 1 comments:

Other Comments for Manuscript:

1. Line 69-80, It would be better to be concise.

We made this section more concise.

2. Line 85-86, “RGCCA” instead of “Regularized and Sparse Gener-alized Canonical Correlation Analysis (RGCCA)”.

We thank the reviewer for the correction and implemented the suggested change (line 83-85).

4. Line 94, “shown in shown in the top left”, twice? 

We thank the reviewer for identified the duplication in the text and corrected it (line 90). 

In Fig 1.tiff, Smaller font of the name of the potato accessions would be better.

We thank the suggestion. Unfortunately, we consider that if the font was further reduced it would not be visible.

5. Line 141, Significant level for p-values is 0.01 or 0.05?

The significant level for p-value is 0,1. thank you for noticing this. We have fixed this error in the manuscript (line 121). 

6. Line 227, In Fig 5, it would be better to scale in the X axis for each chromosome individually, instead of in an accumulated scale for all 12 chromosomes.

We appreciate the suggestion but respectfully disagree because we think that the suggested change would add complexity to Fig 5. Therefore, we did not implement this change.

7. Line 344, “were likely responsible for” instead of “where likely responsible for”?

We made the change and apologize for the error (line 342).

8. Line 375-397, The two sections, “Selection of material to be sequenced” and “RNA extraction and sequencing”, could be merged. Anyway, the sequencing methods could be described in a brief way.

We implemented the suggested change.

9. Line 383, 37 (even 94) accessions are not a large number of accessions, considering the sequencing cost. How the statistical analyses support “one transcriptome per accession”?

We are aware that it would have been ideal to have replicates per genotype and a larger number of sampled accession but we the statistical tests that we perform are robust to these caveats and allow us to address our research questions. Specifically, we conducted differential expression analyses using different accessions of the same coloration as biological replicates (at least 9 replicates per category). Additionally, we conducted correlation and co-expression analyses, which do not require replication per accession and work appropriately with 37 data points. We believe that this is a correct strategy to identify genes that show similar patterns across different accessions with the similar anthocyanin content, which is the goal of our study. Our goal is not to characterize or compare specific accessions and therefore we do not require replication per accession. We also believe that 37 accessions is a sufficiently big population to study the diversity of gene expression and anthocyanin content. Finally, as shown in the manuscript, most of the genes identified in our analyses have previous anthocyanin annotations, which confirms that or strategy is appropriate to identify anthocyanin genes. 

We clarify this in the methods section (line 389-394).

10. The genetic backgrounds of 37 landraces are different, even though all these landraces belong to Phureja group. Did the difference interfere the Differential Expression Analyses? It should be discussed in the manuscript.

When we selected accessions with similar coloration patterns, we selected representative accessions in the different phylogenetic groups (Supplementary Figures 1 and 3). In this way we minimized the effect of population structure in our analyses.

This was clarified in the corrected version of the manuscript (line 374-378).

11. The quality of the written English is not satisfying. There are spelling and vocabulary mistakes needing corrections in the manuscript. Such as “an-thocyanin” in abstract. References in the text need to be enclosed in brackets, such as line 378.

Following the reviewer suggestion, we improved readability and corrected typos and other errors. We apologize for the typos and language errors present in the previous version of the manuscript. In this version of the manuscript, we carefully proofread the text and corrected all the errors highlighted by the editor.

12. There are two names of the MYB gene (Soltu. DM.10G020850), StAN1 and StAN2.

The code: Soltu. DM.10G020850 is just for StAN2 not for StAN1. We corrected this error.

13.The differential genes analyzed were not verified by any experiments, such as qPCR and dual luciferase reporter assay.

The goal of this study was to identify genes that showed population-level correlations between expression and anthocyanin content. Most of the candidate genes identified have already been associated to anthocyanin production in previous studies of potato cultivars. In the future we plan to validate some of the identified candidates with transgenesis and qPCR however we believe it is beyond the scope of this manuscript.

14. The WRKY13 gene in this manuscript has been reported and should be included in the discussion. StWRKY13 promotes anthocyanin biosynthesis in potato (Solanum tuberosum) tubers. Functional Plant Biology, 2022, 49(1):102-114

We appreciate this suggestion and included the suggested reference (line 609-610).

15. “our results are summarized in Fig 4” in line 167 should be deleted.

We were grateful for this recommendation and we removed this section from the paragraph (line 169).

Detailed responses to reviewer # 2 comments: 

Reviewer #2: The manuscript “The contribution of gene expression to anthocyanin diversity in potato landraces (Solanum tuberosum L. Phureja)” is well presented. This can be accepted after addressing minor comments as below:

Title: Modify it, something like: ‘Uncover anthocyanin diversity in potato landraces applying RNA-seq.......approaches’

The title was changed to “Uncovering anthocyanin diversity in potato landraces (Solanum tuberosum L. Phureja) using RNA-seq”.

Abstract: Line 26, modify it, or mention first and second crop. Since there are several other vegetables rich in antioxidants like tomato etc.

The three crops that provide the highest proportion of antioxidants in the human diet are maize, tomato and potato. We have made the respective change in the manuscript.

Introduction: Line 40-41, modify, not good sentence. Modify from L40 to 64, in a story format. Its a bit scattered.

We have made the requested change and those paragraphs of the manuscript have been re-written (line 39-63).

Results: L89-90: Besides two, mention the correlation between phylogeny and anthocyanin production as well.

We are grateful for this recommendation and made the suggested correction (line 88-89).

L102: (ie spots) ?? Correct ‘i.e.’ spots

We were grateful for this recommendation and we made this correction (line 101).

Check for typo and grammatical errors in throughout the manuscript.

Following the reviewer suggestion, we improved readability and corrected typos and other errors. We apologize for the typos and language errors present in the previous version of the manuscript. In this version of the manuscript, we carefully proofread the text and corrected all the errors highlighted by the editor.

---

## [Decision Letter · Decision Letter 1]

19 Aug 2022

Uncovering anthocyanin diversity in potato landraces (Solanum tuberosum L. Phureja) using RNA-seq

PONE-D-22-16485R1

Dear Dr. Roda,

We’re pleased to inform you that your manuscript has been judged scientifically suitable for publication and will be formally accepted for publication once it meets all outstanding technical requirements.

Kind regards,

Pankaj Bhardwaj, Ph.D.

Academic Editor

PLOS ONE

Additional Editor Comments (optional):

Reviewers' comments:

Reviewer's Responses to Questions

**Comments to the Author**

1. If the authors have adequately addressed your comments raised in a previous round of review and you feel that this manuscript is now acceptable for publication, you may indicate that here to bypass the “Comments to the Author” section, enter your conflict of interest statement in the “Confidential to Editor” section, and submit your "Accept" recommendation.

Reviewer #1: All comments have been addressed

Reviewer #2: All comments have been addressed

2. Is the manuscript technically sound, and do the data support the conclusions?

Reviewer #1: Yes

Reviewer #2: Yes

3. Has the statistical analysis been performed appropriately and rigorously? 

Reviewer #1: Yes

Reviewer #2: Yes

4. Have the authors made all data underlying the findings in their manuscript fully available?

Reviewer #1: Yes

Reviewer #2: Yes

5. Is the manuscript presented in an intelligible fashion and written in standard English?

Reviewer #1: Yes

Reviewer #2: Yes

6. Review Comments to the Author

Reviewer #1: The The author answered all the comments. I think this vision is appropriate for publication. I have no more comment.

Reviewer #2: Maybe accepted for publication. Authors have addressed all the comments. Hence the manuscript may be accepted.

7. PLOS authors have the option to publish the peer review history of their article (what does this mean?). If published, this will include your full peer review and any attached files.

Reviewer #1: No

Reviewer #2: **Yes: **Jagesh Kumar Tiwari

---

## [Editor Report · Acceptance letter]

30 Aug 2022

PONE-D-22-16485R1 

Uncovering anthocyanin diversity in potato landraces (*Solanum tuberosum* L. Phureja) using RNA-seq 

Dear Dr. Roda:

I'm pleased to inform you that your manuscript has been deemed suitable for publication in PLOS ONE. Congratulations! Your manuscript is now with our production department. 

Kind regards, 

on behalf of

Dr. Pankaj Bhardwaj 

Academic Editor

PLOS ONE